# Discovery of Ureido-Based Apcin Analogues as Cdc20-specific Inhibitors against Cancer

**DOI:** 10.3390/ph16020304

**Published:** 2023-02-15

**Authors:** Yiqin He, Xiangyang Le, Gaoyun Hu, Qianbin Li, Zhuo Chen

**Affiliations:** 1Department of Medicinal Chemistry, Xiangya School of Pharmaceutical Sciences, Central South University, Changsha 410013, China; 2Hunan Key Laboratory of Organ Fibrosis, Changsha 410013, China; 3Hunan Key Laboratory of Diagnostic and Therapeutic Drug Research for Chronic Diseases, Changsha 410013, China

**Keywords:** Cdc20-specific inhibitor, ureido analogues, Apcin analogues, anticancer, docking

## Abstract

Cdc20 is a promising drug target that plays an important role in the mid-anaphase process of cellular mitosis, and Apcin is the only reported core structure of the Cdc20-specific inhibitor. Some potent Apcin derivatives were obtained in our previous research, and a structure–activity relationship was determined. In this study, we designed and synthesized a series of ureido-based Apcin derivatives. The proliferation-inhibition experiments on four cancer-cell lines showed that ureido skeleton could promote the anti-proliferation activity of purine-substituted compounds, whereas the ureido analogues with pyrimidine substitutes showed no significant improvement in the inhibitory effect compared with the original ones. Further tests confirmed that ureido-based compounds can enhance the binding affinity to Cdc20 by increasing the levels of Cdc20 downstream proteins. Compound **27** revealed a remarkably antitumor activity pattern against Hela (IC_50_ = 0.06 ± 0.02 μM) and potent binding affinity to Cdc20. Moreover, compound **20** induced caspase-dependent apoptosis and cell-cycle arrest at the G2/M phase, and compound **27** induced caspase-dependent apoptosis and promoted microtubule polymerization. Finally, a molecular-docking simulation was performed for compounds **20** and **27** to predict the potential ligand–protein interactions with the active sites of the Cdc20 proteins.

## 1. Introduction

Anaphase-promoting complex/cycle (APC/C) is a multifunctional ubiquitin protein ligase that ubiquitinates against diverse ubiquitinated substrates to regulate various cellular processes such as cell division, differentiation, genome stability, energy metabolism, cell death, autophagy, and carcinogenesis [1,2]. APC/C works only when two coactivators—Cdc20 and Cdh1—join in. Ubiquitination of substrates by APC/C requires the formation of an APC/C-activator-substrate complex, and Cdc20 and Cdh1 act on APC/C to target specific substrates at different stages of the cell cycle. The WD40 domains of Cdc20 and Cdh1 located at C-terminus provide a binding platform to recruit APC/C substrates [3,4,5,6].

During mitosis, the spindle-assembly checkpoint (SAC) induces the generation of the mitotic-checkpoint complex (MCC), which consists of C-Mad2, Cdc20, BubR1, and Bub3, and acts to inactivate APC/CCdc20. When all kinetochores are properly attached to the microtubules, it signals MCC disassembly, which breaks down, thereby activating APC/CCdc20. APC/CCdc20 leads to subsequent ubiquitination and degradation of cyclin B, securin, etc., by 26S protease to allow metaphase-to-anaphase transition [7,8,9,10].

TCGA statistics show that the Cdc20 gene is overexpressed in a variety of human tumor tissues, including breast cancer, cervical cancer, colorectal cancer, liver cancer, and other cancer tissues. Statistical analysis of clinical trials by many researchers provides evidence of the association between high expression of Cdc20 and cancer patients. Cdc20 may be a promising target for the treatment of cancer [11,12,13,14,15]. 

Cdc20 downstream substrates include Bim, cyclin B1, securin, Sox2, etc. Cyclin B1 plays an important role in the process of cell mitosis. At the end of the middle of mitosis, cyclin B1 is rapidly degraded, the chromatids are separated, and cells enter anaphase. Securin degrades and releases segregase, which blocks separase/ESPL1 function during mitosis, preventing proteolysis of the cohesin complex and subsequent chromosome segregation [16,17]. At the onset of anaphase, securin is ubiquitinated, leading to self-destruction and the release of ESPL1. In addition to regulating the process of mitosis, Cdc20 also participates in cell apoptosis by targeting Mcl-1 and Bim [18]. Mcl-1 and Bim are an anti-apoptotic member and pro-apoptotic member of the Bcl-2 protein family, respectively, and play a key role in the apoptosis-signaling pathway. It is shown that depletion of Cdc20 or drug inhibition of APC results in increased apoptosis [19,20,21]. Most evidence suggest that Cdc20 regulates apoptosis by controlling the ubiquitination and stability of the pro-apoptotic protein Bim [22,23].

Apcin was the only reported specific inhibitor of Cdc20 that occupies the WD-40 domain of Cdc20 protein, competitively inhibiting binding of Cdc20 to its downstream substrates. Apcin has poor antiproliferative inhibitory activity, so a series of Apcin derivatives was synthesized in our previous research, in which some potent compounds were obtained, such as **7b**, **7d**, and **9f** [19]. At the same time, we found that **9f** not only possesses an inhibitory effect on Cdc20 but can also inhibit the aggregation of microtubules, realizing a “two-punch strategy” (strong mitotic arrest followed by blocking mitotic exit). Since the ureido might have better chemical stability and biological affinity than carbamate, which is the core scaffold of Apcin, in this study, we designed and synthesized a series of ureido-based Apcin derivates, expecting to enhance their affinity with Cdc20 while promoting anticancer activity. Furthermore, a couple of mechanistic experiments were conducted to screen out potential Cdc20-specific inhibitors.

## 2. Results and Discussion

### 2.1. Chemistry

We synthesized 30 compounds (Table 1) following two schemes. In Figure 1, the starting material of R_1-_NH_2_ was reacted with p-nitrophenyl chloroformate to obtain an intermediate, which was directly ammonolyzed without any further treatment to obtain ureido compound. Ureido compound was condensed with trichloroacetaldehyde hydrate and then chlorinated to obtain an intermediate, which was finally subjected to nucleophilic substitution with R_2_-NH_2_ to obtain the target products. In Figure 2, the isocyanate method for generating the urea group R_3_-NH_2_ was first reacted with hydrated trichloroacetaldehyde, and then chlorination was carried out and an aminolysis intermediate was obtained. Finally, the target product was obtained by reacting isocyanate with the aminolysis intermediate.

### 2.2. In Vitro Antiproliferative Assay and Prediction of Lipid Permeability and Toxicity in Silico

As shown in Table 2, the results of the antiproliferative assay in vitro of the target compounds were similar in different cell lines, among which Mcf-7 and Mda-mb-132 are both breast-cancer cells, HepG2 is a hepatocellular carcinoma cell, and Hela is a cervical-cancer cell. Among all compounds, purine derivative 27 revealed a remarkably broad antitumor activity pattern against Hela (IC_50_ = 0.06 ± 0.02 μM), MCF-7 (IC_50_ = 0.27 ± 0.06 μM), MDA-MB-231 (IC_50_ = 0.32 ± 0.04 μM), and Hepg2 (IC_50_ = 0.24 ± 0.11 μM), which was better than that of compound **9f** obtained in the previous study. Dichloro-substituted pyrimidine derivative 20 showed potent antiproliferative effects against all the tested tumor-cell lines and had stronger proliferation-inhibition activity compared to that of the previously synthesized pyrimidine compound **7d** (IC_50_ = 63.20 ± 0.90 μM, Hela). At the same time, the lower cLogP value of compound **20** compared to compound **7d** implies that compound **20** had greater lipid permeability; in addition, the toxicity value of compound **20** was slightly less than that of compound **7d** (Table 3).

Among the ureido-based Apcin derivatives, the proliferation-inhibition experiments on four cancer-cell lines showed that the ureido analogues with pyrimidines substitutes had no significant alteration to the original ones (**1** vs. Apcin vs. **20** vs. **19**), whereas the ureido skeleton could promote the activity of the purine-substituted compound (**27** vs. **9f**). Compound **9f** had greater lipid permeability than compound **27** (Table 3), suggesting that compound **27** has stronger anti-proliferative activity due to other mechanisms. Compounds with no substituents on pyrimidines were described as R_2_-NH_2_ (Figure 1), including nitroimidazole (1), phenethyl (2), cyclohexyl (3), 2-morpholinoethylamine (4), ethanolamine (5), and benzenesulfonyl (6), all of which showed poor activity (IC_50_ > 80 μM). Besides, the substitution on the pyrimidines displayed a strong influence on activity, such as chlorine (9), cyano (15), fluorine (16), which showed dramatically reduced activities versus the most active compound, whereas in the pyrimidine substituted with electron-withdrawing substituents, such as trifluoromethyl (7), meta-dichloro (17), and dichloroamino (20), the antiproliferative activity of the compound was greatly enhanced (IC_50_ < 40μM). In addition, the position of the substituents on the pyrimidine also had some effect on the activity. When only the amidopyrimidine counterpoint was replaced, the activity of the compounds was very poor (**9**, **10**, **13**, **14**, **15**). When the amidopyrimidine was substituted in the ortho- and meso- positions, the activity of the compounds was significantly enhanced (**7**, **20**, **22**–**26**). Multi-substituted compounds like compounds **20**, **25**, and **26** showed good activity compared with Apcin.

Under the premise of the ureido-based Apcin derivatives, the activity of 2-fluoroadenine compounds against Hela (27, IC_50_ = 0.06 ± 0.02 μM; 28, IC_50_ = 0.17 ± 0.08 μM; 29, IC_50_ = 0.08 ± 0.01 μM) was much stronger than that of any pyrimidine compounds. Among the purine compounds, the activity of compound **27** (IC_50_ = 0.06 ± 0.02 μM, Hela) substituted with 2-morpholinoethylamine in R_3_-NH_2_ was stronger than that of phenethylamine-substituted compound **28** (IC_50_ = 0.17 ± 0.08 μM). It was speculated that the difference in water solubility might be the reason for the difference in activity of the compounds.

In summary, the data on toxicity (Table 3) showed that most of the compounds may have had hepatotoxicity, reproductive toxicity, and acute oral toxicity, but the acute oral toxicity was relatively small. In addition, a few compounds may have had nephrotoxicity. Ureido-based Apcin analogues exhibited better antiproliferative activity compared to carbamate-based ones. The ureido-based pyrimidine compounds exhibited the most effective antiproliferative activity when metronidazole was substituted with a benzene ring, and the substituent activity was phenethyl > metronidazole > 2-morpholineethanol against Hela. However, morpholine was the preferred structural element of the purine compounds (2-morpholineethanol > metronidazole > phenethyl), which is consistent with our previous findings (9f, IC_50_ = 0.3 ± 0.2 μM; 9a, IC_50_ > 300 μM; 9b, IC_50_ = 51.2 ± 0.9 μM; Hela) [19]. We speculate that it is because the membrane permeability of benzyl-substituted purine compounds was too poor to enter the cell for their efficacy. However, the predicted results of cLogP values (Table 3) were inconsistent with the results of anti-proliferation experiments (28 > 27 > 29, metronidazole > 2-morpholineethanol > phenethyl). 

When R_3_-NH_2_ was substituted with morpholine, ureido-based purine compound **27** (IC_50_ = 0.06 ± 0.02 μM, Hela) showed at least 3000-fold more activity than the Apcin reference compound (IC_50_ = 181.88 ± 12.49 μM, Hela). Fluoro-substituted and ureido-based analogues exhibited the most effective antiproliferative activity. Based on the results of the Western blot, as shown below(Figure 1, the level of Cdc20 protein in Hela cells was relatively high, which was selected as the cell line for further experiments. Therefore, we selected pyrimidine compound **20** and purine compound **27** as the compounds with the strongest proliferation-inhibition activity against Hela for subsequent mechanism research.

### 2.3. Surface Plasmon Resonance (SPR) Assay

Surface plasmon resonance (SPR) assay provides information on the affinity and kinetics of molecular interactions, and the affinity value (Kd) associated with the interaction may be used to investigate the binding efficiency. The results of the SPR experiment are shown in Table 4 and Figure 2. Compounds **20** and **27** exhibited higher binding ability to Cdc20 protein than compounds **7d** and **9f**.

Interaction of compounds **20** and **27** with Cdc20, along with Apcin acting as the reference compound, were determined by SPR experiments. The results showed that compounds **20** and **27** and Apcin interacted with the human recombinant Cdc20 protein on the chip, with Kd values of 79.6 μM, 97.0 μM, and 236 μM, respectively (as shown in Figure 2 and Table 4). The experimental results prove that the two selected compounds were consistent with Apcin and could be reversibly combined with Cdc20, with obvious binding and dissociation phases. Among them, the binding affinity of compound **20** to Cdc20 (Kd = 79.6 μM) was about three times that of Apcin and Cdc20 (Kd = 236 μM) and was slightly stronger than that of the previous compound **7d**. As compound **20** showed potent antiproliferative effects against all the tested tumor-cell lines compared to compound **7d**, it is suggested that the binding affinity of compounds **20** and **7d** was consistent with its antiproliferative potency.

Compound **27**, a compound representative of purine, showed a stronger affinity with Cdc20 (Kd = 97 μM) than Apcin (Kd = 236 μM), and was higher than the previous compound **9f**. Considering the results of purine compound **27** and pyrimidine compound **20**, the binding affinity was inconsistent with its antiproliferative potency (Kd value: 20 > 27 > Apcin, IC_50_ value: 27 < 20 < Apcin), suggesting that compound **27** may have other anti-cancer mechanisms. The results prove that ureido-based Apcin analogues may be more advantageous than carbamate ones.

### 2.4. Molecular-Docking Simulation 

In this study, we simulated the interaction of the compound with the Cdc20 protein (PDB ID: 4n14) using Molecular Operating Environment (MOE 2015) and showed the ligand interactions between compound and protein through the 2D map in Figure 3with the corresponding score. The results of the affinity values showed that ureido-based Apcin analogues had stronger binding ability with Cdc20 than the carbamate ones (20 > 27 > 7d > 9f).

All of the compounds formed hydrogen bonds with backbone atoms from Asp177. The hydrophobic trichloromethyl groups of compounds **20** and **27** were found to be buried in the pocket, which is similar to Apcin. When compound **7d** was compared with compound **20**, compound **20** increased one hydrogen interaction with Gly214 (Figure 3B,3C), and the absolute binding score of compound **20** with Cdc20 protein was greater than that of compound **7d** (**20** vs. **7d**), which proved that compound **20** had stronger binding ability with Cdc20 protein. The docking results are consistent with the results of our SPR experiment. In addition, compared with compound **9f**, compound **27** did not show much advantage in molecular docking (Figure 3D,3E). They only showed two hydrogen bonds with Asp177; however, their absolute binding score with Cdc20 protein shows that compound **27** had stronger binding ability with Cdc20 (**27** vs. **9f**), which explains the result of the SPR experiment. Due to the restriction of the Cdc20 protein pocket, the ureido was exposed outside the pocket and was unable to form hydrogen bonds with the surrounding amino acids. We suspect that compound **27** had an unknown interaction with Cdc20 protein and may need to be elucidated by eutectic structure with Cdc20 protein. Our next plan is to consider exploring the binding mode of Cdc20 protein and compound **27**.

All docking results show that ureido-based Apcin analogues could maintain a similar binding mode as Cdc20 binding with Apcin and had stronger binding ability with Cdc20 than the carbamate ones.

### 2.5. Western Blot

The results of the Western blot show that compounds **20** and **27** increased the levels of Cdc20 downstream proteins but had no effect on the Cdc20 protein level, consistent with Apcin (Figure 1 and Figure 4).

It is reported that Cdc20 protein is overexpressed in various cancer-cell lines. We selected four cells lines, including Mda-mb-231, Hepg2, Mcf-7, and Hela, to investigate the Cdc20 levels and detect the expression of Cdc20 (Figure 1). The results show that the level of Cdc20 protein in Hela cells was relatively high, and was selected as the cell line for further experiments. Similarly, neither compound **20** nor compound **27** had any effect on the level of Cdc20 protein in Hela cells, as shown in Figure 4, indicating that the compounds did not act by regulating the translational and post-translational modification of Cdc20, consistent with the mechanism of action of a Cdc20-specific inhibitor.

Cyclin B1, securin, and Bim are the key specific downstream substrates of Cdc20, which are related to cell cycle and apoptosis. Cyclin B1 is mainly expressed in the G2/M phase of cells and regulates the cell-cycle progression. Securin protein prevents the proteolysis of the cohesin complex and the subsequent segregation of the chromosomes during mitosis. As shown in Figure 4, after the cells were treated with compound **20** or **27**, the level of cyclin B1 and securin protein was significantly increased, which is consistent with Apcin. Cleaved PARP is a 116 a nuclear polymerase, which is a highly conserved ribozyme involved in DNA repair and apoptosis. Bim is a protein in the Bcl-2 family that has pro-apoptotic activity. Both compound **20** and compound **27** could significantly increase the expression of cleaved PARP and Bim, and the effect of compound **27** was more significant, indicating that the pro-apoptotic effect of compound **27** is stronger than that of compound **20**. These results reveal that compounds **20** and **27** were consistent with the mechanism of Apcin as Cdc20 inhibitors. At the same time, further experiments are needed to verify whether the compounds have any effect on cell cycle and apoptosis.

### 2.6. The Annexin V-FITC/PI Double-Staining Fluorescence Experiment 

The annexin V-FITC/PI double-staining fluorescence experiment of Hela cells was carried out to examine the effect of the compounds on cell apoptosis. As shown in Figure 5 and Figure 6, compounds **20** and **27** induced cell apoptosis and compound **20** blocked cells in the G2/M phase. 

Apoptotic rates were increased in a concentration-dependent way in Hela cells treated with Apcin, compound **20**, and compound **27** for 24 h. Compound **20** at 80 μM and compound **27** at 1 μM showed more potency than Apcin at 300 μM. The results are consistent with the trend of cell proliferation-inhibition experiments and Western blot experiments, indicating that the inhibitory effect of the compound on the proliferation of Hela was probably related to its apoptosis-inducing effect.

To test whether the compound blocked mitotic exit in Hela cells, an annexin V-FITC/PI assay was conducted in Hela cells treated with Apcin, compound **20**, or compound **27** for 24 h, respectively. A significant increase in the number of cells in the mitotic G2/M phase was found after treatment with Apcin or compound **20** (Figure 6), and the experimental results are concentration-dependent: 34.87% of cells were in the G2/M phase after Apcin (150 μM) treatment, and 41.09% of cells were in the G2/M phase after compound **20** (40 μM) treatment, indicating that compound **20** showed stronger ability to block mitotic exit than Apcin did. After treated with compound **27**, the number of cells in the G2/M phase was slightly reduced, but increasing the concentration of 27 had little effect on the cell mitotic exit, which is inconsistent with the results of Apcin. It is suggested that compound **27** may have different mechanisms of pro-apoptotic action than Apcin.

### 2.7. Microtubule Polymerization-Inhibition Experiment 

Our research previously proved that adenine compound **9f** had dual inhibitory effects of Cdc20 and microtubules. Compound **27** can also disrupt the polymerization of tubulin with a different mechanism, as shown in Figure 7.

The structure of compound **27** is similar to that of compound **9f**; therefore, paclitaxel was used as the reference compound to investigate whether compound **27** could disrupt the organization of the cellular microtubule network. A microtubule polymerization-inhibition experiment was carried out in vitro, as shown in Figure 7. Compound **20** at 30 μM slightly promoted microtubule polymerization, but compound **27** at 3 μM and 30 μM was much more effective and promoted microtubule polymerization in a concentration-dependent manner, indicating that compound **27** is a potent promotor of tubulin assembly. Compound **27** and paclitaxel had the same mechanism of action, which could promote microtubule polymerization. Like paclitaxel, compound **27** might prevent the formation of normal mitotic spindles, cause chromosome breakage, and inhibit cell replication, which might be the reason why the cancer-cell proliferation activity of compound **27** was greatly enhanced. 

### 2.8. Bioled-Egg Model Analysis 

We carried out Bioled-egg model analysis using the SwissADME website. The dots in the white ellipse represent compounds that are most likely to be passively absorbed by the gastrointestinal tract. The dots in the yellow represent compounds that are most likely to penetrate the CNS through the BBB. The white and the yolk are not mutually exclusive. Molecules that are not expected to be absorbed well and BBB-permeable molecules are located in the gray area. The results (Figure 8) show that Apcin and compound **27** were substrates of P-GP (drug resistance), and compound **20** was not a substrate of P-GP; it was difficult for the gastrointestinal tract to absorb Apcin, but compounds **20** and **27** were absorbable. None of the three compounds could cross the blood–brain barrier.

In summary, some researchers found that Apcin reduced the proportion of mitotic cells in a dose-dependent manner and shortened the duration of mitosis after nocodazole (microtubule-destabilizing agent) or taxol (microtubule stabilizer) treatment, which was the result of Apcin’s specific binding to the D-box pocket of Cdc20 [24]. The Apcin derivatives we synthesized might have similar microtubule-disrupting effects as nocodazole or taxol, which is also the reason for the greatly increased antiproliferation-inhibition activity. It was suggested that another mechanism of action might exist in compound **27**; further exploration needs to be done.

## 3. Materials and Methods

### 3.1. Chemicals

All purchased reagents and raw materials were of analytical grade (AR) grade and were used directly without further purification. Nuclear magnetic-resonance (NMR) spectroscopy was carried out on a Bruker AVANCEIII-400 and an AVANCEIII-500 NMR. Compounds were dissolved in DMSO-*d_6_* or CDCl_3_, tetramethylsilane (TMS) was used as internal control, chemical shifts (δ) were expressed in parts per million (ppm), coupling constants were expressed in Hertz (Hz), and multiplicity was described as singlet (s), doublet (d), triplet (t), quadruplet (q), multiplet (m), and broad (br). High-resolution mass spectra (HRMS) were recorded using MALDI-TOF-MS/MS (Agilent). Solvent peaks were used as reference values with CDCl_3_ at 7.26 ppm for ^1^H NMR and 77.16 ppm for ^13^C NMR, with DMSO-*d_6_* at 2.50 and 3.33 ppm for ^1^H NMR and 39.52 ppm for ^13^C NMR. High-performance liquid-chromatography (HPLC) analysis of all final compounds was conducted on a Shimadzu 20AT Series HPLC with an ZORBAX Extend–C18 column (5 μm, 100Å, 4.6 × 250 mm, Agilent). The mobile phase was methanol–water, acetonitrile–water, methanol–water (0.05 mol/L NH_4_Ac), or acetonitrile–water (0.05 mol/L NH_4_Ac); the flow rate was 1 mL/min; and the detection wavelength (λ) was 254 nano. All final compounds for biological evaluation were analyzed to achieve a minimum of 95% purity. Compounds were isolated and purified by column chromatography using 200–300-mesh silica gel. The reaction process was monitored by thin-layer chromatography (TLC) using pre-coated silica-gel plates (GF254) with a thickness of 0.25 mm under a UV lamp at a wavelength of 254 nm. For details of Nuclear magnetic-resonance (NMR) and High-resolution mass spectra (HRMS) of our compounds, please download the Appendix A.

#### 3.1.1. General Procedure for the Synthesis of the Final Compounds **1**–**17**

Taking compound **2** as an example, using 2-aminopyrimidine (2.00 g, 21.05 mmol) as the raw material, 10 equivalents of hydrated trichloroacetaldehyde (34.53 g, 0.21 mol) were added, and the reaction was stirred overnight at 100 °C. The white solid intermediate was obtained by recrystallization from ethyl acetate (40 mL), and thionyl chloride (1.50 mL, 20.16 mmol) was added in dry DCM (20 mL) and stirred for 2 h at 40 °C, removing the excess thionyl chloride to obtain the compound with electrophilic chlorine atoms. Aqueous ammonia with a concentration of 25% (4.00 mL, 26.00 mmol) was added dropwise at low temperature and stirred for 2 h, and the intermediate 2,2,2-trichloro-N-(pyrimidin-2-yl)ethane-1,1-diamine (4.79 g, 20.08 mmol) was obtained after adding methanol dropwise. In addition, solid phosgene (19.5 g, 66.12 mmol) and 1 equivalent of phenethylamine (8.00 g, 66.12 mmol) were reacted in 1,4-dioxane (100 mL) solution under reflux stirring at 100 °C overnight, and the reaction solution was cooled at room temperature. Then, the intermediate 2,2,2-trichloro-N-(pyrimidin-2-yl)ethane-1,1-diamine (4.79 g, 20.08 mmol) was added, and the reaction was refluxed and stirred at 100 °C for 4 h. The final product 2 (7.49 g, 19.35 mmol) was obtained by precipitation or purification.

1-(2-(2-methyl-5-nitro-1H-imidazol-1-yl)ethyl)-3-(2,2,2-trichloro-1-(pyrimidin-2-ylamino)ethyl)urea (1). The reaction was performed according to the general procedure. Compound **1** was obtained as a white powder. Yield: 93.4%. HPLC purity: 98.36%. ^1^H NMR (500 MHz, DMSO- *d_6_*): δ = 8.38 (d, *J* = 14.3 Hz, 2H), 8.20 (s, 1H), 7.98 (s, 1H), 7.86 (s, 1H), 6.54 (d, *J* = 15.5 Hz, 2H), 4.29 (d, *J* = 15.4 Hz, 2H), 3.48 (d, *J* = 20.8 Hz, 2H), 2.36–2.30 (m, 3H). ^13^C NMR (101 MHz, DMSO- *d_6_*): δ = 161.36, 158.44, 156.55, 152.02, 138.85, 133.59, 112.66, 110.56, 103.74, 46.70, 46.22, 14.24 ppm. HRMS-ESI (*m*/*z*) calculated for C_13_H_15_Cl_3_N_8_O_3_^+^ [M+H]^+^ 437.0411, found 437.0410.

1-phenethyl-3-(2,2,2-trichloro-1-(pyrimidin-2-ylamino)ethyl)urea (2). The reaction was performed according to the general procedure. Compound **2** was obtained as a white powder. Yield: 91.9%. HPLC purity: 98.25%. ^1^H NMR (400 MHz, DMSO- *d_6_*) δ = 8.39 (d, *J* = 4.8 Hz, 2H), 8.01 (s, 1H), 7.27 (d, *J* = 7.2 Hz, 2H), 7.20 (d, *J* = 7.4 Hz, 3H), 6.90 (s, 1H), 6.78 (s, 1H), 6.69 (d, *J* = 12.4 Hz, 2H), 3.25 (d, *J* = 6.0 Hz, 2H), 2.72–2.64 (m, 2H). ^13^C NMR (126 MHz, DMSO- *d_6_*) δ= 161.45, 158.49, 156.42, 139.93, 129.12, 128.75, 126.49, 112.57, 104.07, 68.73, 66.81, 41.32, 36.23 ppm. HRMS-ESI (*m*/*z*) calculated for C_15_H_16_Cl_3_N_5_O^+^ [M+H]^+^ 388.0499, found 388.0494.

1-cyclohexyl-3-(2,2,2-trichloro-1-(pyrimidin-2-ylamino)ethyl)urea (3). The reaction was performed according to the general procedure. Compound **3** was obtained as a white powder. Yield: 87.0%. HPLC purity: 98.12%. ^1^H NMR (500 MHz, DMSO- *d_6_*) δ = 8.39 (d, *J* = 4.7 Hz, 2H), 8.06 (s, 1H), 6.89 (s, 1H), 6.75 (s, 1H), 6.61 (s, 1H), 6.53 (s, 1H), 3.37 (s, 1H), 1.77–1.45 (m, 5H), 1.32–1.01 (m, 5H). ^13^C NMR (126 MHz, DMSO- *d_6_*) δ = 161.49, 158.46, 155.64, 112.50, 104.18, 68.66, 48.34, 33.54, 33.43, 25.68, 24.76 ppm. HRMS-ESI (*m*/*z*) calculated for C_13_H_18_Cl_3_N_5_O^+^ [M+H]^+^ 366.0655, found 366.0654.

1-(2-morpholinoethyl)-3-(2,2,2-trichloro-1-(pyrimidin-2ylamino)ethyl)urea (4). The reaction was performed according to the general procedure. Compound **4** was obtained as a white powder. Yield: 82.8%. HPLC purity: 98.75%. ^1^H NMR (500 MHz, DMSO- *d_6_*) δ = 8.39 (d, *J* = 4.8 Hz, 2H), 7.98 (s, 1H), 6.90 (s, 1H), 6.83 (s, 1H), 6.76 (s, 1H), 6.60 (s, 1H), 3.56 (d, *J* = 8.7 Hz, 4H), 3.13 (d, *J* = 9.5 Hz, 2H), 2.31 (t, *J* = 6.2 Hz, 6H). ^13^C NMR (101 MHz, DMSO- *d_6_*) δ = 161.47, 158.48, 156.43, 112.55, 104.06, 68.78, 66.57, 58.41, 53.67, 36.72 ppm. HRMS-ESI (*m*/*z*) calculated for C_13_H_20_Cl_3_N_6_O_2_^+^ [M+H]^+^ 397.0713, found 397.0714.

1-ethyl-3-(2,2,2-trichloro-1-(pyrimidin-2-ylamino)ethyl)urea (5). The reaction was performed according to the general procedure. Compound **5** was obtained as a white powder. Yield: 88.4%. HPLC purity: 99.08%. ^1^H NMR (400 MHz, DMSO- *d_6_*) δ = 8.39 (d, *J* = 4.8 Hz, 2H), 8.01 (s, 1H), 7.31–7.16 (m, 5H), 6.90 (s, 1H), 6.75 (s, 1H), 6.69 (d, *J* = 13.5 Hz, 2H), 3.26 (d, *J* = 6.0 Hz, 2H), 2.70 (d, *J* = 7.1 Hz, 2H). ^13^C NMR (101 MHz, DMSO- *d_6_*) δ = 161.44, 158.47, 156.30, 112.56, 104.14, 68.73, 34.59, 15.84 ppm. HRMS-ESI (*m*/*z*) calculated for C_9_H_12_Cl_3_N_5_ONa^+^ [M+Na]^+^ 334.0005, found 334.0006.

N-((2,2,2-trichloro-1-(pyrimidin-2-ylamino)ethyl)carbamoyl)benzenesulfonamide(6). The reaction was performed according to the general procedure. Compound **6** was obtained as a white powder. Yield: 85.4%. HPLC purity: 98.43%. ^1^H NMR (500 MHz, DMSO- *d_6_*) δ = 11.21 (s, 1H), 8.38 (d, *J* = 4.8 Hz, 2H), 8.27 (s, 1H), 7.91 (d, *J* = 7.7 Hz, 2H), 7.72–7.56 (m, 3H), 7.31 (s, 1H), 6.78 (d, *J* = 14.6 Hz, 2H). ^13^C NMR (126 MHz, DMSO- *d_6_*) δ = 161.15, 158.62, 150.85, 139.97, 134.03, 129.62, 127.68, 113.04, 102.71, 68.19 ppm. HRMS-ESI (*m*/*z*) calculated for C_13_H_13_Cl_3_N_5_O_3_S^+^ [M+H]^+^ 423.9802, found 423.9805.

1-phenethyl-3-(2,2,2-trichloro-1-((4-(trifluoromethyl)pyrimidin-2-yl)aminno16)ethyl)urea (7). The reaction was performed according to the general procedure. Compound **7** was obtained as white powder. Yield: 85.9%. HPLC purity: 98.66%. ^1^H NMR (500 MHz, DMSO- *d_6_*) δ = 8.82 (d, *J* = 51.8 Hz, 2H), 7.27 (d, *J* = 7.4 Hz, 2H), 7.19 (dd, *J* = 11.9, 7.8 Hz, 4H), 6.94 (s, 1H), 6.73 (d, *J* = 17.7 Hz, 2H), 3.27 (d, *J* = 5.9 Hz, 2H), 2.68 (d, *J* = 7.2 Hz, 2H). ^13^C NMR (101 MHz, DMSO- *d_6_*) δ = 162.05, 161.74, 156.36, 139.89, 129.12, 128.74, 126.48, 122.34, 119.61, 107.45, 103.43, 68.88, 41.26, 36.21 ppm. HRMS-ESI (*m*/*z*) calculated for C_16_H_15_Cl_3_F_3_N_5_NaO^+^ [M+Na]^+^ 478.0192, found 478.0187.

4-methyl-N-((2,2,2-trichloro-1-((4-(trifluoromethyl)pyrimidin-2-yl)amino)ethyl)carbamoyl)benzenesulfonamide (8). The reaction was performed according to the general procedure. Compound **8** was obtained as a white powder. Yield: 92.4%. HPLC purity: 99.54%. ^1^H NMR (500 MHz, DMSO- *d_6_*) δ = 11.14 (s, 1H), 9.06 (s, 1H), 8.76 (s, 1H), 7.78 (s, 2H), 7.40 (d, *J* = 7.8 Hz, 2H), 7.33 (s, 1H), 7.25 (s, 1H), 6.76 (s, 1H), 2.37 (s, 3H). ^13^C NMR (101 MHz, DMSO- *d_6_*) δ = 162.07, 161.45, 150.90, 144.54, 137.09, 129.98, 127.75, 122.22, 119.48, 107.98, 107.95, 102.13, 68.24, 21.43 ppm. HRMS-ESI (*m*/*z*) calculated for C_15_H_14_Cl_3_F_3_N_5_O_3_S^+^[M+H]^+^ 505.9835, found 505.9835.

1-phenethyl-3-(2,2,2-trichloro-1-((5-chloropyrimidin-2-yl)amino)ethyl)urea (9). The reaction was performed according to the general procedure. Compound **9** was obtained as white powder. Yield: 94.6%, HPLC purity: 98.66%. ^1^H NMR (500 MHz, DMSO- *d_6_*) δ = 8.48 (s, 2H), 8.38 (d, *J* = 8.6 Hz, 1H), 7.23 (dd, *J* = 34.6, 7.5 Hz, 5H), 6.82 (s, 1H), 6.69 (d, *J* = 16.2 Hz, 2H), 3.26 (d, *J* = 6.0 Hz, 2H), 2.69 (d, *J* = 12.4 Hz, 2H). ^13^C NMR (126 MHz, DMSO- *d_6_*) δ = 159.93, 156.61, 156.37, 139.91, 129.12, 128.75, 126.49, 119.83, 103.67, 69.14, 41.29, 36.22 ppm. HRMS-ESI (*m*/*z*) calculated for C_15_H_16_Cl_4_N_5_O^+^ [M+H]^+^ 422.0109, found 422.0605.

N-((2,2,2-trichloro-1-((5-chloropyrimidin-2yl)amino)ethyl)carbamoyl)benzenesul fonamide (10). The reaction was performed according to the general procedure. Compound **10** was obtained as a white powder. Yield: 93.2%. HPLC purity: 99.15%. ^1^H NMR (500 MHz, DMSO- *d_6_*) δ = 11.21 (s, 1H), 8.57 (s, 1H), 8.48 (s, 2H), 7.91 (d, *J* = 7.5 Hz, 2H), 7.72–7.56 (m, 3H), 7.32 (s, 1H), 6.68 (s, 1H). ^13^C NMR (101 MHz, DMSO- *d_6_*) δ = 159.59, 156.81, 150.87, 139.94, 134.04, 129.63, 127.70, 120.40, 102.34, 68.55 ppm. HRMS-ESI (*m*/*z*) calculated for C_13_H_12_Cl_4_N_5_O_3_S^+^ [M+H]^+^ 457.9419, found 457.9415.

N-((2,2,2-trichloro-1-((4,6-dichloro-5-methylpyrimidin-2yl)amino)ethyl)carbamoyl)benzenesulfonamide (11). The reaction was performed according to the general procedure. Compound **11** was obtained as a white powder, Yield: 89.7%, HPLC purity: 98.25%. ^1^H NMR (500 MHz, DMSO- *d_6_*) δ = 11.15 (s, 1H), 9.00 (s, 1H), 7.91 (d, *J* = 7.7 Hz, 2H), 7.69 (s, 1H), 7.63 (d, *J* = 7.5 Hz, 2H), 7.30 (s, 1H), 6.48 (s, 1H), 2.23 (s, 3H). ^13^C NMR (101 MHz, DMSO- *d_6_*) δ = 158.29, 150.82, 139.85, 134.08, 129.65, 127.71, 117.45, 101.85, 68.43, 15.35 ppm. HRMS-ESI (*m*/*z*) calculated for C_14_H_12_Cl_5_N_5_NaO_3_S^+^ [M+Na]^+^ 527.9005, found 527.9001.

1-phenethyl-3-(2,2,2-trichloro-1-((4,6-dichloro-5-methylpyrimidin-2-yl)amino)ethyl)urea (12). The reaction was performed according to the general procedure. Compound **12** was obtained as a white powder. Yield: 89.8%, HPLC purity: 99.12%. ^1^H NMR (400 MHz, DMSO- *d_6_*) δ = 8.86 (s, 1H), 7.29 (d, *J* = 12.4 Hz, 2H), 7.24–7.16 (m, 3H), 6.77 (s, 1H), 6.67 (d, *J* = 11.2 Hz, 2H), 3.25 (d, *J* = 5.3 Hz, 2H), 2.71 (s, 2H), 2.25 (s, 3H). ^13^C NMR (101 MHz, DMSO- *d_6_*) δ = 161.31, 158.63, 156.31, 139.87, 129.11, 128.74, 126.48, 116.61, 103.16, 69.10, 41.25, 36.22, 15.32 ppm. HRMS-ESI (*m*/*z*) calculated for C_16_H_17_Cl_5_N_5_O^+^ [M+H]^+^ 469.9875, found 469.988.

1-propyl-3-(2,2,2-trichloro-1-((5-nitropyrimidin-2-yl)amino)ethyl)urea (13). The reaction was performed according to the general procedure. Compound **13** was obtained as white powder, Yield: 89.6%, HPLC purity: 98.55%. ^1^H NMR (400 MHz, DMSO- *d_6_*) δ = 9.63 (s, 1H), 9.24 (s, 1H), 9.18 (s, 1H), 6.95 (s, 1H), 6.74 (d, *J* = 19.0 Hz, 2H), 2.97 (d, *J* = 7.3 Hz, 2H), 2.50 (s, 3H), 1.38 (d, *J* = 7.3 Hz, 2H). ^13^C NMR (126 MHz, DMSO- *d_6_*) δ = 162.84, 156.26, 155.94, 155.39, 135.90, 102.76, 69.25, 41.53, 23.35, 11.77 ppm. HRMS-ESI (*m*/*z*) calculated for C_10_H_13_Cl_3_N_6_NaO_3_^+^ [M+Na]^+^ 393.0012, found 393.0029.

1-phenethyl-3-(2,2,2-trichloro-1-((5-nitropyrimidin-2-yl)amin-o)ethyl)urea (14). The reaction was performed according to the general procedure. Compound **14** was obtained as a white powder. Yield: 91.2%, HPLC purity: 98.91%. ^1^H NMR (400 MHz, DMSO- *d_6_*) δ = 9.24 (s, 1H), 9.17 (s, 1H), 7.28 (d, *J* = 7.2 Hz, 2H), 7.21 (d, *J* = 7.5 Hz, 4H), 7.00 (s, 1H), 6.87 (s, 1H), 6.75 (s, 1H), 3.28 (d, *J* = 5.7 Hz, 2H), 2.69 (d, *J* = 7.3 Hz, 2H). ^13^C NMR (126 MHz, DMSO- *d_6_*) δ = 162.83, 156.26, 155.95, 155.39, 139.86, 135.87, 129.12, 128.76, 126.51, 102.66, 69.21, 41.26, 36.17 ppm. HRMS-ESI (*m*/*z*) calculated for C_15_H_15_Cl_3_N_6_NaO_3_^+^ [M+Na]^+^ 455.0171, found 455.0169.

1-phenethyl-3-(2,2,2-trichloro-1-((5-cyanopyrimidin-2-yl)amin-o)ethyl)urea (15). The reaction was performed according to the general procedure. Compound **15** was obtained as a white powder. Yield: 80.1%, HPLC purity: 98.52%. ^1^H NMR (500 MHz, DMSO- *d_6_*) δ = 9.19 (s, 1H), 8.82 (s, 2H), 7.28 (d, *J* = 14.8 Hz, 3H), 7.20 (d, *J* = 7.4 Hz, 3H), 6.91 (s, 1H), 6.83 (s, 1H), 6.76 (s, 1H), 3.27 (d, *J* = 12.9 Hz, 3H), 2.68 (d, *J* = 15.9 Hz, 2H). ^13^C NMR (101 MHz, DMSO- *d_6_*) δ = 162.35, 161.99, 161.52, 156.32, 139.86, 129.12, 128.76, 126.51, 126.41, 117.00, 102.90, 98.08, 68.76, 41.37, 41.27, 39.60, 36.65, 36.17, 22.96 ppm. HRMS-ESI (*m*/*z*) calculated for C_16_H_15_Cl_3_N_6_NaO^+^ [M+Na]^+^ 435.0271, found 435.0264.

1-phenethyl-3-(2,2,2-trichloro-1-((5-fluoropyrimidin-2-yl)amin-no)ethyl)urea(16). The reaction was performed according to the general procedure. Compound **16** was obtained as a white powder. Yield: 85.2%, HPLC purity: 99.23%. ^1^H NMR (400 MHz, DMSO- *d_6_*) δ = 8.49 (s, 2H), 8.11 (s, 1H), 7.27 (d, *J* = 6.9 Hz, 3H), 7.20 (d, *J* = 7.3 Hz, 3H), 6.81 (s, 1H), 6.72 (d, *J* = 18.8 Hz, 2H), 3.25 (d, *J* = 12.8 Hz, 2H), 2.68 (d, *J* = 14.7 Hz, 2H). ^13^C NMR (126 MHz, DMSO- *d_6_*) δ = 158.40, 156.52, 153.84, 151.88, 139.88, 129.10, 128.73, 126.47, 103.89, 69.51, 41.34, 36.21 ppm. HRMS-ESI (*m*/*z*) calculated for C_15_H_15_Cl_3_FN_5_NaO^+^[M+Na]^+^ 370.1083, found 370.1086.

1-phenethyl-3-(2,2,2-trichloro-1-((2,6-dichloropyrimidin-4-yl)amino)ethyl)urea (17). The reaction was performed according to the general procedure. Compound **17** was obtained as a white powder. Yield: 49.5%, HPLC purity: 99.10%. ^1^H NMR (500 MHz, DMSO- *d_6_*) δ = 8.98 (s, 1H), 7.30 (d, *J* = 7.4 Hz, 2H), 7.21 (d, *J* = 7.6 Hz, 3H), 6.99 (s, 1H), 6.79 (d, *J* = 15.2 Hz, 2H), 6.46 (s, 1H), 3.29 (d, *J* = 10.5 Hz, 2H), 2.70 (d, *J* = 5.1 Hz, 2H). ^13^C NMR (126 MHz, DMSO- *d_6_*) δ = 164.07, 159.10, 158.73, 156.23, 139.80, 129.11, 128.73, 126.48, 103.65, 102.29, 67.94, 41.19, 36.19 ppm. HRMS-ESI (*m*/*z*) calculated for C_15_H_14_Cl_5_N_5_NaO^+^ [M+Na]^+^ 477.9539, found 479.9509.

#### 3.1.2. General Procedure for the Synthesis of the Final Compounds **18**–**30**

The synthesis of compounds was reported earlier by our research group [19]. Taking compound **20** as an example, 1.2 equivalents of p-nitrophenyl chloroformate (3.98 g, 19.80 mmol) were added to phenylethylamine (2.00 g, 16.50 mmol), dichloromethane (20 mL) was used as a solvent, and an appropriate amount of triethylamine was added to neutralize the hydrochloric acid produced by the reaction, and the reaction solution was directly cooled at low temperature. After stir for 1 h, aqueous ammonia (2.50 mL, 16.25 mmol) was added dropwise to the trap, stirred for 3 h, and the intermediate was obtained by adding methanol to precipitate. The intermediate and chloral hydrate (16.54 g, 0.10 mol) were stirred overnight and reacted at 100 °C, and ethyl acetate (40 mL) was added to separate out a white solid. Then, thionyl chloride (1.50 mL, 20.17 mmol) was added to reflux for 2 h and stirred to remove excess chloride. After the sulfone, an intermediate substituted with a nucleophile was obtained, which was stirred with 4,6-dichloro-1,5-diaminopyrimidine (3.00 g, 16.75 mmol) in dry THF (30 mL) at 50 °C for 6 h to obtain a crude product, which was separated by column chromatography to obtain the target compound **20** (4.43 g, 9.43 mmol) as a white solid.

1-phenethyl-3-(2,2,2-trichloro-1-((4,6-dichloropyrimidin-2-yl)amino)ethyl)urea, sodium salt (18). The reaction was performed according to the general procedure. Compound **18** was obtained as a white powder. Yield: 45.0%, HPLC purity: 99.56%. ^1^H NMR (500 MHz, DMSO- *d_6_*) δ = 9.18 (s, 1H), 7.31–7.19 (m, 5H), 7.15 (s, 1H), 6.82 (s, 1H), 6.70 (d, *J* = 16.0 Hz, 2H), 3.28 (d, *J* = 5.9 Hz, 2H), 2.73–2.64 (m, 2H). ^13^C NMR (101 MHz, DMSO- *d_6_*) δ = 161.66, 161.07, 156.30, 139.85, 129.11, 128.74, 126.48, 110.51, 102.89, 69.03, 41.24, 36.22 ppm. HRMS-ESI (*m*/*z*) calculated for C_15_H_14_Cl_5_N_5_NaO^+^ [M+Na]^+^ 477.9539, found 477.9538.

Phenethyl(1-((5-amino-4,6-dichloropyrimidin-2-yl)amino)-2,2,2-trichloroethyl)carbamate (19). The reaction was performed according to the general procedure. Compound **19** was obtained as a white powder. Yield: 65.7%, HPLC purity: 98.12%. ^1^H NMR (400 MHz, DMSO- *d_6_*) δ = 7.89 (s, 1H), 7.33–7.14 (m, 6H), 6.33 (s, 1H), 5.15 (s, 2H), 4.38–4.14 (m, 2H), 2.88 (d, *J* = 13.5 Hz, 2H). ^13^C NMR (126 MHz, DMSO- *d_6_*) δ = 155.71, 151.25, 145.68, 138.32, 129.56, 129.28, 128.72, 126.76, 126.72, 102.83, 70.92, 65.84, 35.10 ppm. HRMS-ESI (*m*/*z*) calculated for C_15_H_14_Cl_5_N_5_NaO_2_^+^ [M+Na]^+^ 493.9488, found 493.9483.

1-(1-((5-amino-4,6-dichloropyrimidin-2-yl)amino)-2,2,2-trichloroethyl)-3-phenethylurea (20). The reaction was performed according to the general procedure. Compound **20** was obtained as a white powder. Yield: 57.2%, HPLC purity: 98.25%. ^1^H NMR (500 MHz, DMSO-*d_6_*) δ = 8.03 (s, 1H), 7.31–7.16 (m, 5H), 6.65 (d, *J* = 11.4 Hz, 2H), 6.55 (s, 1H), 5.06 (s, 2H), 3.25 (d, *J* = 7.0 Hz, 2H), 2.68 (d, *J* = 14.9 Hz, 2H). ^13^C NMR (101 MHz, DMSO-*d_6_*) δ = 156.40, 151.98, 145.64, 139.90, 129.12, 128.91, 128.74, 126.48, 103.78, 69.49, 41.27, 36.24 ppm. HRMS-ESI (*m*/*z*) calculated for C_18_H_19_F_3_NO_2_S^+^ [M+H]^+^ 370.1083, found 370.1086.

1-(1-((5-amino-4,6-dichloropyrimidin-2-yl)-l4-azanyl)-2,2,2-trichloroethyl)-3-(2-morpholinoethyl)urea (21). The reaction was performed according to the general procedure. Compound **21** was obtained as a white powder. Yield: 54.6%, HPLC purity: 98.33%. ^1^H NMR (500 MHz, DMSO-*d_6_*) δ = 8.00 (s, 1H), 6.82 (s, 1H), 6.57 (d, *J* = 13.8 Hz, 2H), 5.04 (s, 2H), 3.57 (s, 5H), 3.14 (s, 2H), 2.34 (d, *J* = 18.5 Hz, 6H). ^13^C NMR (101 MHz, DMSO-*d_6_*) δ = 156.49, 152.00, 145.64, 128.87, 103.80, 70.23, 69.59, 66.61, 58.40, 53.69, 36.76 ppm. HRMS-ESI (*m*/*z*) calculated for C_13_H_19_Cl_5_N_7_O_2_^+^ [M+H]^+^ 480.0042, found 480.0045.

1-phenethyl-3-(2,2,2-trichloro-1-((4,6-difluoropyrimidin-2-yl)amino)ethyl)urea(22). The reaction was performed according to the general procedure. Compound **22** was obtained as a white powder. Yield: 89.0%, HPLC purity: 98.65%. ^1^H NMR (500 MHz, DMSO-*d_6_*) δ = 9.14 (s, 1H), 7.29 (d, *J* = 15.0 Hz, 2H), 7.24–7.17 (m, 3H), 6.80 (s, 1H), 6.69 (d, *J* = 9.6 Hz, 2H), 6.46 (s, 1H), 3.28 (d, *J* = 5.9 Hz, 2H), 2.70 (d, *J* = 5.7 Hz, 2H). ^13^C NMR (101 MHz, DMSO-*d_6_*) δ = 173.57, 171.15, 161.17, 156.31, 139.84, 129.11, 128.74, 126.49, 102.75, 81.95, 69.09, 41.24, 36.17 ppm. HRMS-ESI (*m*/*z*) calculated for C_15_H_14_Cl_3_F_2_N_5_NaO^+^ [M+Na]^+^ 446.0130, found 446.0133.

1-phenethyl-3-(2,2,2-trichloro-1-((5-methylpyrimidin-2-yl)ami-no)ethyl)urea (23). The reaction was performed according to the general procedure. Compound **23** was obtained as a white powder. Yield: 65.7%, HPLC purity: 98.75%. ^1^H NMR (500 MHz, DMSO-*d_6_*) δ = 8.24 (s, 1H), 7.87 (s, 1H), 7.31–7.15 (m, 5H), 6.90 (s, 1H), 6.71–6.62 (m, 3H), 3.25 (d, *J* = 7.0 Hz, 2H), 2.69 (d, *J* = 11.9 Hz, 2H), 2.30 (s, 3H). ^13^C NMR (126 MHz, DMSO-*d_6_*) δ = 161.32, 156.42, 139.94, 129.11, 128.74, 126.47, 111.96, 104.24, 68.75, 41.34, 36.26, 24.07 ppm. HRMS-ESI (*m*/*z*) calculated for C_16_H_19_Cl_3_N_5_O^+^ [M+H]^+^ 402.0654, found 402.0655.

Methyl-4-methyl-2-((2,2,2-trichloro-1-(3-phenethylureido)eth-yl)amino)pyrimidine-5-carboxylate (24). The reaction was performed according to the general procedure. Compound **24** was obtained as a white powder. Yield: 60.6%,HPLC purity: 98.46%. ^1^H NMR (500 MHz, DMSO-*d_6_*) δ = 8.78 (d, *J* = 17.9 Hz, 2H), 7.45–7.13 (m, 5H), 6.97 (t, *J* = 9.5 Hz, 1H), 6.87–6.67 (m, 2H), 4.26 (q, *J* = 7.1 Hz, 2H), 3.27 (q, *J* = 6.8 Hz, 2H), 2.69 (td, *J* = 7.0, 2.5 Hz, 2H), 2.60 (d, *J* = 8.4 Hz, 3H), 1.30 (t, *J* = 7.1 Hz, 3H). ^13^C NMR (126 MHz, DMSO-*d_6_*) δ = 164.99, 161.82, 160.77, 156.32, 139.90, 129.11, 128.73, 126.47, 114.03, 103.47, 68.75, 60.76, 41.29, 25.17, 24.48, 14.58 ppm. HRMS-ESI (*m*/*z*) calculated for C_18_H_23_Cl_3_N_6_O_3_^+^ [M+NH_3_]^+^ 476.0897, found 476.0836.

1-(1-((2-amino-6-chloropyrimidin-4-yl)amino)-2,2,2-trichloroe-thyl)-3-phenethylurea(25). The reaction was performed according to the general procedure. Compound **25** was obtained as a white powder. Yield: 25.3%, HPLC purity: 98.36%. ^1^H NMR (500 MHz, DMSO-*d_6_*) δ = 7.80 (s, 1H), 7.31–7.16 (m, 5H), 7.05–6.70 (m, 3H), 6.65 (s, 1H), 6.61 (s, 1H), 5.85 (s, 1H), 3.32–3.19 (m, 2H), 2.69 (d, *J* = 11.1 Hz, 2H). ^13^C NMR (126 MHz, DMSO-*d_6_*) δ = 165.49, 161.23, 158.35, 156.32, 139.95, 129.12, 128.75, 126.48, 104.16, 68.67, 41.33, 36.25 ppm. HRMS-ESI (*m*/*z*) calculated for C_15_H_17_Cl_4_N_6_O ^+^ [M+H]^+^ 437.0218, found 437.0226.

1-(1-((4-amino-6-chloropyrimidin-2-yl)amino)-2,2,2-trichloroe-thyl)-3-phenethylurea (26). The reaction was performed according to the general procedure. Compound **26** was obtained as a white powder. Yield: 28.8%, HPLC purity: 98.99%. ^1^H NMR (500 MHz, DMSO-*d_6_*) δ = 7.86 (s, 1H), 7.33–7.18 (m, 5H), 6.89 (s, 1H), 6.78 (s, 1H), 6.67 (s, 2H), 6.46 (s, 1H), 5.98 (s, 1H), 3.27 (d, *J* = 6.8 Hz, 2H), 2.70 (d, *J* = 6.9 Hz, 2H). ^13^C NMR (126 MHz, DMSO-*d_6_*) δ = 163.54, 163.08, 158.68, 156.25, 139.91, 129.14, 128.77, 126.51, 103.56, 93.29, 67.38, 41.21, 36.23 ppm. HRMS-ESI (*m*/*z*) calculated for C_15_H_17_Cl_4_N_6_O ^+^ [M+H]^+^ 437.0218, found 437.0226.

1-(1-(6-amino-2-fluoro-9H-purin-9-yl)-2,2,2-trichloroethyl)-3-(2-morpholinoethyl)urea (27). The reaction was performed according to the general procedure. Compound **27** was obtained as a light-yellow powder. Yield: 45.8%, HPLC purity: 98.90%. ^1^H NMR (400 MHz, DMSO-*d_6_*) δ = 8.34 (s, 1H), 8.19 (s, 1H), 8.01 (d, *J* = 25.5 Hz, 2H), 6.94 (s, 1H), 6.49 (s, 1H), 3.69–3.45 (m, 6H), 2.39–2.30 (m, 6H). ^13^C NMR (101 MHz, DMSO-*d_6_*) δ = 160.56, 158.08, 155.91, 151.59, 139.21, 116.66, 99.80, 70.85, 66.46, 58.12, 53.61, 36.59 ppm. HRMS-ESI (*m*/*z*) calculated for C_14_H_19_Cl_3_FN_8_O_2_^+^ [M+H]^+^ 455.0681, found 455.1061.

1-(1-(6-amino-2-fluoro-9H-purin-9-yl)-2,2,2-trichloroethyl)-3-phenethylurea (28). The reaction was performed according to the general procedure. Compound **28** was obtained as a white powder. Yield: 49.7%, HPLC purity: 98.38%. ^1^H NMR (500 MHz, DMSO-*d_6_*) δ = 8.33 (s, 1H), 8.00 (d, *J* = 45.7 Hz, 2H), 7.27 (d, *J* = 23.5 Hz, 2H), 7.18 (t, *J* = 7.4 Hz, 3H), 6.94 (s, 1H), 6.57 (s, 1H), 3.28 (d, *J* = 6.2 Hz, 2H), 2.69 (d, *J* = 18.6 Hz, 2H). ^13^C NMR (126 MHz, DMSO-*d_6_*) δ = 158.28, 155.92, 151.66, 151.50, 139.70, 139.26, 129.10, 128.76, 126.53, 116.68, 99.87, 70.87, 41.36, 35.99 ppm. HRMS-ESI (*m*/*z*) calculated for C_16_H_16_Cl_3_FN_7_O^+^ [M+H]^+^ 446.0466, found 446.0125.

1-(1-(6-amino-2-fluoro-9H-purin-9-yl)-2,2,2-trichloroethyl)-3-(2-(2-methyl-5-nitro-1H-imidazol-1-yl)ethyl)urea (29). The reaction was performed according to the general procedure. Compound **29** was obtained as a yellow powder. Yield: 32.7%, HPLC purity: 98.31%. ^1^H NMR (500 MHz, DMSO-*d_6_*) δ = 8.33 (s, 1H), 8.22 (d, J = 10.1 Hz, 1H), 7.94 (s, 3H), 6.84 (d, J = 10.1 Hz, 1H), 6.69 (s, 1H), 4.31 (h, *J* = 8.7, 8.1 Hz, 2H), 3.47 (ddt, *J* = 23.6, 17.6, 5.8 Hz, 2H), 2.27 (s, 3H). ^13^C NMR (126 MHz, DMSO-*d_6_*) δ = 160.06, 158.29, 157.27, 156.10, 151.84, 138.94, 133.48, 116.65, 99.69, 89.02, 70.79, 56.53, 46.51, 14.15 ppm. HRMS-ESI (*m*/*z*) calculated for C_23_H_23_F_3_N_3_O_2_S^+^ [M+H]^+^ 495.0376, found 495.0378.

1-phenethyl-3-(2,2,2-trichloro-1-(2-(4-(trifluoromethyl) pyrimi-din-2-yl)hydrazinyl)ethyl)urea (30). The reaction was performed according to the general procedure. Compound **30** was obtained as a yellow powder. Yield: 67.0%, HPLC purity: 98.38%. ^1^H NMR (400 MHz, DMSO-*d_6_*) δ = 9.27 (s, 1H), 8.69 (s, 1H), 7.44–7.03 (m, 6H), 6.51 (s, 1H), 6.38 (s, 1H), 5.60 (d, *J* = 44.1 Hz, 2H), 3.25–3.09 (m, 2H), 2.69–2.54 (m, 2H). ^13^C NMR (101 MHz, DMSO-*d_6_*) δ = 163.95, 161.95, 157.14, 139.94, 129.10, 128.74, 126.46, 106.73, 101.96, 75.12, 41.20, 36.29 ppm. HRMS-ESI (*m*/*z*) calculated for C_16_H_17_Cl_3_F_3_N_6_O^+^ [M+H]^+^ 471.0482, found 471.0478.

### 3.2. Cell Culture and Cytotoxicity Assay

We used four different cell lines: human cervical-cancer cells (Hela), human 1H breast-cancer cells (Mda-Mb-132), human breast-cancer cells (Mcf-7), and human liver-cancer cells (HepG2). Hela and MDA-MB-132 cells were grown in DMEM (Gibco) containing 10% FBS and 1% double antibody (penicillin 100 U/mL, streptomycin 100 μg/mL, Solarbio). MCF-7 and HepG2 cells were grown in 1640 (Gibco) containing 10% FBS and 1% double antibody (penicillin 100 U/mL, streptomycin 100 μg/mL, Solarbio). All cell lines were purchased from the Xiangya Cell Bank, Central South University, Changsha, China, and incubated at 37 °C with 5% CO2 in a humidified atmosphere. When the cell density reached more than 80%, the cells were passaged at a ratio of 1:2. Cytotoxicity assay was assessed using CCK-8 methods. The cells were seeded in a 96-well plate (2 × 10^3^–3 × 10^3^ cells/well), the 96-well plate was incubated at 37 °C with 5% CO_2_ in a humidified atmosphere, and the cells were grown in a monolayer; the original culture was discarded. A total of 100 μL of drug-containing medium with a specific concentration gradient was added and then cultivated for 48 h; 10 μL/well of CCK-8 solution was added, put in an incubator, and incubated for 1 h and read with a multi-function microplate reader at a wavelength of 450 nm. The inhibition rate was calculated, and the IC_50_ results were calculated by GraphPad Prism 7 software. The experiment was repeated three times.

### 3.3. Surface Plasmon Resonance Analysis

Surface plasmon resonance (SPR) analysis can measure the interaction process between various biomolecules, such as polypeptides, proteins, oligonucleotides, and oligosaccharides, as well as viruses, bacteria, cells, and small molecular compounds. First, we installed the COOH chip according to the standard operating procedure of the OpenSPRTM instrument and started running it at the maximum flow rate (150 μL/min). The detection buffer was PBS, and after reaching the signal baseline, 200 μL of isopropanol were loaded and the air was removed by running for 10 s. After reaching baseline, the sample loop was flushed with buffer and evacuated with air, and after the signal reached baseline, the buffer flow rate was adjusted to 20 μL/min. A total of 200 μL of EDC/NHS (1:1) solution (20 μL/min, 4 min), 200 μL EDC/NHS (1:1) solution (20 μL/min, 4 min), and 200 μL recombinant human Cdc20 protein buffer was loaded and run for 4 min (20 μL/min), and the sample loop was rinsed with buffer and drained with null air. A total of 200 μL of blocking solution was loaded (20 μL/min, 4 min), and the sample loop was flushed with buffer and evacuated with air. After the baseline was stable, the analyte was diluted with buffer and loaded at 20 μL/min. The binding time of the analyte and the ligand was 240 s, and the natural dissociation time was 480 s. The analysis software used for the experimental results was TraceDrawer (Ridgeview Instruments ab, Sweden), and the analysis method was the one-to-one analysis model.

### 3.4. Western Blot Assay

The lysate for extracting cellular proteins was prepared with bromophenol blue (0.02%), dithiothreitol (DTT; 0.5M), glycerol (30%), sodium dodecyl sulfate (SDS; 10%) and Tris-Cl (0.25M, pH 6.8). Protein extracts were separated according to molecular weight by the polyacrylamide gel electrophoresis (PAGE) method and then transferred to solid-phase support (PVDF membrane). The PVDF membrane was placed in a blocking solution containing 5% skim milk for 2 h, followed by incubation with primary and secondary antibodies, and the bands were detected and imaged in the ChemiDocTMXRS+ imaging system.

### 3.5. Cell-Apoptosis Assay

The Hela cells were seeded in a 6-well plate (2 × 10^5^–3 × 10^5^ cells/well), the 6-well plate was incubated at 37 °C with 5% CO_2_ in a humidified atmosphere, and the cells were grown in a monolayer. Then, the preset drug concentration was added and incubated for 24 h. Cells were collected by centrifugation, resuspended in buffer, and then stained by adding Annexin V-FITC (5 μL) and PI Staining Solution (5 μL) and incubated at room temperature for 10 min. The samples were detected by flow cytometry within 1 h after staining. Flow cytometer (Beckman Coulter Cytofex) was used for subsequent detection, and the data were processed with FlowJo software.

### 3.6. Cell-Cycle Assay

The Hela cells were seeded in a 6-well plate (2 × 10^5^–3 × 10^5^ cells/well), the 6-well plate was incubated at 37 °C with 5% CO_2_ in a humidified atmosphere, and the cells were grown in a monolayer. Then, the preset drug concentration was added and incubated for 24 h. Cells were collected by centrifugation and were fixed overnight with pre-cooled ethanol (70%) and washed twice with buffer, propidium-iodide staining solution was added (0.5 mL/well), and they were incubated at 37 °C for 30 min in the dark for flow detection.

### 3.7. Tubulin Polymerization Assay In Vitro

We used an HTS-Tubulin Polymerization Assay Kit (BK004P, Cytoskeleton, USA) to assess the inhibitory effect of compounds on tubulin polymerization. Tubulin solution was prepared, and the compound solution and paclitaxel solution were tested according to the instructions of the kit. General tubulin buffer, paclitaxel solution, and 10× test compound solution were added to a 96-well plate (10 μL/well) and incubated at 37 °C for two minutes. A total of 100 μL of tubulin solution was pipetted into the corresponding wells. The plate was immediately placed on a multi-function microplate reader at 37 °C, and the continuous kinetic values were recorded.

### 3.8. Molecular Docking

The crystal structure of Cdc20 in complex with different ligands was downloaded from PDB, which was previously used to determine the Apcin-binding site of Cdc20 [25]. (http://www.rcsb.org/, accessed on 6 May 2021; PDB codes 4n14). Missing hydrogen atoms in the crystal structure were computationally added; proteins were preprocessed by 3D protonation, Mg^2+^, GDP, and GTP; and all the other bound small molecules, except the target ligand, were deleted. Ligand structures were built with MOE.2015 and minimized using the MMFF94x force field. The ligands were then prepared to generate low-energy ring conformers. Molecular docking was performed using MOE due to the ability of molecules to bind to the ligand sites [26].

## 4. Conclusions

In this study, we found that ureido skeleton could promote the anti-proliferation activity of purine-substituted compounds, and the ureido-based Apcin derivatives had stronger binding ability to Cdc20 than the carbamate-based structure. Compound **27** emerged as Cdc20 inhibitor is valuable in cancer treatment. Compounds **27** deserves further exploration of its anticancer mechanism.

## Data Availability

Electronic supplementary information (ESI) available: Molecular model of compound binding to Cdc20 protein; Copies of the ^1^H and ^13^C NMR spectra for all molecules; Data graph of biological experiment in the article.

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
