# Peer review of "Discovery of Ureido-Based Apcin Analogues as Cdc20-specific Inhibitors against Cancer"

_pharmaceuticals, 2023, doi:10.3390/ph16020304_

Round 1

Reviewer 1 Report

The work presented by Chen and co-workers describes the synthesis of heterocyclic compounds to be evaluated as antiproliferative agents. The proliferative experiments showed that the ureido compounds possess anti-proliferative activity, while the ureido analogs had no significant improvement compared to the original ones.

These results can be helpful to the scientific community, nonetheless, it is suggested  the next points improve the quality of the presentation:

1. Line 87, says vitro must say in vitro (in italics)

2. Line 122, The authors mention that the results of antiproliferative activity were contrary to their previous findings, here, it is suggested to explain these phenomena in terms of the Structure-Activity Relationship (SAR).

3. Line 158-169. Please, also discuss these results using the affinity values obtained from docking.

Line 243. Scheme 1. Find a better way to present the scheme synthesis of these compounds. Honestly, is quite confusing to follow as is presented, also, including the yield and reaction time of each reaction.  For instance, the first reaction could be indicated as R-NH2 instead of Part B, and the same for part A.

Line 251. Table 1. Please chose one way to represent the compounds and homogenize them. Sometimes the CF3 is represented as the carbon attached to the F by single bonds and other it is represented by CF3.

Lines 331 and 332. The number ins this formula NH4Ac must be subscript.

Line 338. General procedure. Even though you are using a general procedure, it is recommended to indicate the quantity in g, mg, mL, etc, of each reactant. 

Line 347, 350. Indicate the units of the temperature

Line 338. Section 4.1.1. The general section contains several mistakes, for instance, the coupling constant should be written in italics, and the number in HRMS formulas should be in subscripts, also including the error in ppm.

The 13C ran in 500 Mhz is reported as (101 MHz) and should be (125 MHz), also de indicate that the solvent is deuterated. 

The yields must be reported using one decimal place.

All these mistakes are from lines 338 to 594.

Line 482. Section 4.1.2. Indicate the quantities used in the general procedure, for instance, how much DCM and Et3N were used. Please detail it as much as possible.

I suggest accepting the manuscript after the minor revision.

Reviewer 2 Report

This manuscripts is interesting and authors have investigated in-vitro activity along with some in-silico calculations.

# Please provide proper referencing in the docking methodology. I find no referencing there.

# Please go through these literature and do cite them concerning your methodology. https://doi.org/10.1016/j.medidd.2019.100008; https://doi.org/10.3390/ASEC2021-11157; https://doi.org/10.2174/1573409916666200615141047

# I found no properly written spectral data. Authors have forgotten where cations or ionic charges should be mentioned in molecular ions of corresponding mass analysis of compounds. 

# Same follows for all spectral data. 

# Why authors have chosen only PDB codes 4n14? Please cite why this pdb id the right chose?

# I suggest authors to please atleast add molecular dynamics analysis of best in-vitro or best docked compound with selected target chosen. at least 20 ns simulation should be added.

# Authors needs to calculate in-silico ADMET data and added in Table form.

# Please calculate Bioled-egg model analysis using SwissADME website.

# Please calculate various in-silico toxicity aspects of all compounds using 'admetSAR' website.

# Docking section lacks proper elaboration. Do correct accordingly.

Reviewer 3 Report

1.     In section 2.2 in vitro antiproliferative assay: a) Page 2 lines 93-96, according to the structure, compound 20 is more closely related to compound 7b in your previous study instead of 7d. b) On Page 3 lines 104-109, the SAR analysis is not sufficient, some compounds are multi-substituted, and additional compounds (like compound 14) should also be included for analysis. In addition, the substitution position should also be considered in the SAR analysis. c) On Page 3, lines 110-112, please specify the cell line that is related to the IC50, and if it is related to the Hela cell line, then the IC50 of compound 29 should be 0.08. d) On Page 3, lines 120-122, please specify the compound IDs and add the previous results for clarification. e) Page 3, lines 126-128, if only based on the anti-proliferation activity, then compound 20 should not be chosen for further study. Please clarify the reasons for choosing compound 20 for further study. f) In table 2, please also add the IC50 values for compounds 7b and 7d from your previous study.

2.     Since the study is based on the previous study, other than using Apcin as the control, please also include compound 9f in the SPR, western blot, apoptosis, cell cycle arrest, and tubulin polymerization assays.

3.     There is a deviation between the Cdc20 binding affinity and anti-proliferation activity of compound 27. May compound 27 have some off-target effects?

4.     Please add the quantitative illustration for the western blot results.

5.     There are many typos in the current manuscript (e.g., page 4, line 159-165, fig1 should be fig2), please make a thorough revision and polish the language as well.

Round 2

Reviewer 2 Report

Authors have incorporated suggestions

Reviewer 3 Report

The authors have revised the manuscript based on the reviewers' comments. I would recommend 'Accept in present form'.